# Influence of the COVID-19 Pandemic on Adherence to Orally Administered Antineoplastics

**DOI:** 10.3390/jcm11092436

**Published:** 2022-04-26

**Authors:** Amparo Talens, Elsa López-Pintor, Marta Bejerano, Mercedes Guilabert, María Teresa Aznar, Ignacio Aznar-Lou, Blanca Lumbreras

**Affiliations:** 1Servicio de Farmacia, Hospital General Universitario de Elda, 03600 Alicante, Spain; talens_amp@gva.es (A.T.); bejarano_mar@gva.es (M.B.); 2Departamento de Ingeniería, Área de Farmacia y Tecnología Farmacéutica, Universidad Miguel Hernández, 03550 Alicante, Spain; elsa.lopez@umh.es; 3Center for Biomedical Research in Epidemiology and Public Health Network (CIBERESP), 28029 Madrid, Spain; ignacio.aznar@sjd.es; 4Departamento de Psicología de la Salud, Universidad Miguel Hernández, 03202 Alicante, Spain; mguilabert@umh.es; 5Servicio de Farmacia, Hospital Universitario San Juan de Alicante, 03550 Alicante, Spain; aznar_mte@gva.es; 6Teaching, Research & Innovation Unit, Institut de Recerca Sant Joan de Déu (IRSJD), 08830 Barcelona, Spain; 7Departamento de Salud Pública, Historia de la Ciencia y Ginecología, Universidad Miguel Hernández, 03550 Alicante, Spain

**Keywords:** COVID, adherence, orally administered antineoplastics, haemato-oncology patients

## Abstract

Background: Several factors can influence adherence to orally administered antineoplastics, including fear or anxiety resulting from situations such as the COVID-19 pandemic. The aim of this study was to analyse the influence of these patients’ experiences on adherence to orally administered antineoplastics. Methods: Cross-sectional study in four hospitals including >18 year old cancer patients receiving orally administered antineoplastics during the first half of 2021. Data were collected from medical records and through telephone interviews. Adherence was assessed through the prescription refill records and pill counts. Patients’ fear resulting from the pandemic was assessed by means of a structured questionnaire using a 5-point Likert-type scale. Results: Our sample compr BARCELONAised 268 patients (54% men) with a mean age of 64 years (SD 12). More than 15% had experienced afraid and 5% had experienced a dangerous situation when attending hospital, 17% felt they had received less care, and 30% preferred telepharmacy. Adherence measured by pill count was 69.3% and 95.5% according to prescription refill records. Patients who had experienced fear or anxiety when attending hospital were less adherent (aOR 0.47, 95% CI 0.23–0.96, *p* = 0.039). Conclusion: The fear experienced by some patients has affected adherence to treatment.

## 1. Introduction

The COVID-19 pandemic has increased the vulnerability of cancer patients. Their immunosuppressed status caused by both the disease and the therapy, puts them at greater risk of SARS-CoV-2 infection [1] and complications that could lead to ICU admission and mechanical ventilation [2,3]. Several studies described higher COVID mortality rates in cancer patients than in the general population, which vary according to the type, stage and spread of the neoplasm [4]. For this reason, various scientific societies, such as the European Society for Medical Oncology [5], the American Society of Clinical Oncology [6], the National Comprehensive Cancer Network [7] and Cancer Care Ontario [8], have established diagnostic and management guidelines aimed at preventing contagion by prioritising telemedicine over outpatient care, the prioritisation of telemedicine whenever it is appropriate because it does not require physical examination or administration of non-delayable treatments and accessibility is possible and technically feasible [9].

The COVID-19 pandemic has dominated healthcare system priorities, limiting the care received by people with chronic diseases such as cancer [10]. This situation, combined with patients’ fear of becoming infected, has hindered routine health care [10]. Sarid N et al. [11] demonstrated how the adoption of certain measures by the medical team (communication with the patient and changes in the care routine) decreased the anxiety of cancer patients. In addition, Karacin C et al. [12] observed that patients who delayed their scheduled hospital visits, both for treatment and routine tests, did so for fear of contracting COVID-19. The percentage of patients delaying their appointments decreased with the introduction of telemedicine.

The pandemic has therefore affected important aspects of medical treatment, such as adherence. However, the literature examining the influence of COVID-19 on adherence to treatment is inconclusive and varies according to the disease under study. Kaye et al., evaluated adherence to inhalers in patients with asthma and chronic obstructive pulmonary disease showing an increase during the pandemic [13]. In people living with HIV, however, adherence seems to have decreased [14]. Studies including patients with rheumatic diseases undergoing treatment with immunosuppressants found no reduction in adherence. However, among patients who did stop taking their medication, the most frequently cited reason for non-adherence was fear of increased risk of infection when attending the hospital to collect their medication [15,16].

The responsibility of complying with orally administered antineoplastics plans lies with the patients themselves. Under normal healthcare practice conditions, the prevalence of adherence ranges from 46% to 100%, depending on the drugs prescribed and the method used to measure adherence [17]. Several factors influence adherence orally administered antineoplastics, including the fear or anxiety resulting from situations such as the COVID-19 pandemic. To date, however, no published studies have assessed the association between these feelings of fear and/or anxiety and adherence to orally administered antineoplastics during the COVID-19 pandemic. Previous studies have focused on the influence of the COVID-19 pandemic on adherence to antineoplastics including patients with different profiles. For example, in a previous study, Sarid N et al. [11] only included haematological cancer patients. Karacin C et al. [12] included patients on parenteral treatment. Patients on orally administered antineoplastics come to the hospital for medical consultation and to collect their treatment from the pharmacy but they did not usually stay for a long period in the hospital in contact with other patients, so the experience is different from those patients on parenteral treatment. Given that due to the pandemic COVID-19, the available recommendations established to substitute intravenous therapies for orally administered antineoplastics whenever it was appropriate [18], assessing these patients’ opinions, experiences, and preferences, as well as the impact of the COVID-19 pandemic on their adherence, was relevant.

Establishing a profile of those patients most affected by the pandemic could help us to establish future improvements that will ensure the success of pharmacological treatments. Our hypothesis was based on evidence that certain clinical and sociodemographic variables affect patients’ susceptibility to fear of contracting COVID-19 during a hospital visit and, consequently, their adherence to treatment.

This study aimed to determine which cancer patient characteristics were associated with greater fear regarding health care during the COVID-19 pandemic, and to what extent this fear influenced their adherence to treatment.

## 2. Materials and Methods

### 2.1. Design and Scope

We performed a cross-sectional study in adult cancer patients receiving orally administered antineoplastics from outpatient pharmaceutical care units in four general and university hospitals in the province of Alicante in the southeast of Spain. The coordinating centre was Elda General University Hospital, which has 548 beds and serves a geographically dispersed population of 189,629 inhabitants. Alicante General University Hospital is the tertiary care centre of the province and the largest of the four participating sites, with 841 beds and a catchment population of 280,535 inhabitants. San Juan General University Hospital and Elche General University Hospital have 407 and 448 beds, respectively, and catchment populations of 225,153 and 169,599 inhabitants, respectively [19].

Clinical trial registration: NCT04550533 (https://clinicaltrials.gov/ct2/show/NCT04550533?term=NCT04550533&draw=2&rank=1, accessed on 24 April 2022)

### 2.2. Participants

#### 2.2.1. Inclusion Criteria

We included cancer patients aged 18 years and older who were taking orally administered antineoplastics from the Anatomical Therapeutic Chemical class L01 or L02, and who collected their treatment from outpatient pharmaceutical care units. (Appendix A) Patients who were unable to answer the survey or who had been in treatment for less than three months were excluded.

#### 2.2.2. Sample Size Calculation

A previous study analysed the relationship between the impact of the COVID pandemic and adherence to treatment strategies and diagnostic follow-up in patients treated for blood cancer and showed that between 73% and 81% of patients attended their scheduled visits [11]. Taking the mean value of this range, with a confidence level of 95% and 5% precision, we calculated that we would have to include 267 patients.

#### 2.2.3. Selection Method

Participants were recruited in person when collecting their medication from the participating hospitals (when they had an in-person appointment with the specialist or when starting or changing chemotherapy treatment). We recruited patients consecutively March–June 2021 until we reached the established sample size.

During the recruitment visit, patients were provided with information on the study and were invited to participate by the pharmacist responsible for the study in each hospital. All patients who agreed and who met the inclusion criteria were then asked to sign an informed consent form, and a date was set for the telephone interview. Data were collected during these interviews and from patients’ electronic medical records. Researchers from the four hospitals participating in the study were previously trained through two training sessions on the objectives and methodology of the study and the patient interview technique.

### 2.3. Variables

The primary outcome variable was adherence to treatment, which we assessed using two instruments: (a) prescription refill records, when patients picked up their medication or called to have it sent to their home; (b) pill counts (calculated by subtracting the count of the number of pills remaining from the total number of pills dispensed divided by the period (in days) multiplied by 100%). In view of the high adherence to antineoplastic agents reported in previous studies (16), we defined adequate adherence as a pill count above 90%. This level is stricter than the commonly used 80% to 90% [20] but similar to that used by other authors [21].

The secondary outcome was the existence of fear during the pandemic, assessed by means of a structured questionnaire including the following questions: (a) Have you felt afraid when attending hospital?; (b) Have you experienced any dangerous situations when attending hospital?; (c) Have you felt that non-COVID patients have received less care?; and (d) Do you prefer remote pharmaceutical care (i.e., having your medication sent to you rather than picking it up from the hospital)? These questions were included in a survey that was designed specifically for this study by a group of experts through a Delphi procedure. Patients answered the survey questions using a 5-point Likert-type scale, with responses ranging from “never” to “always”. The responses were divided into two groups for the statistical analysis: never, almost never and sometimes versus almost always and always.

The independent variables collected included sociodemographic characteristics: sex, age, living situation (living with family; under 65 and living alone; 65 or over and living alone), educational attainment (no schooling; primary education; secondary or university education), work status (unemployed; in work; retired); clinical characteristics: diagnosis, department the patient was attending (oncology or haematology), time since diagnosis, disease stage, ECOG Performance Status; and characteristics of the treatment: type of treatment (continuous or in cycles), time on current treatment, time on orally administered antineoplastics, total time on chemotherapy treatment, treatment objective (palliative or adjuvant), presence of adverse effects, number of daily doses, type of initial chemotherapy (oral or intravenous).

### 2.4. Statistical Analysis

We estimated both, the prevalence of adherence to treatment (primary outcome) and the existence of fear, measured through the structured questionnaire (secondary outcomes), and their variation according to relevant variables. To compare each category with selected patients’ characteristics, χ^2^ test or Fisher’s exact test was used.

Finally, we estimated the relationship (OR and 95% CIs) between the primary and the secondary outcomes and the variables included in the study through an unconditional logistic regression. After analysing possible interactions between variables and performing all possible two-way tests, the final multivariable model considered all variables that were significant in the univariate analyses (*p* < 0.05) and used forward stepwise selection.

All analyses were carried out with the statistical software IBM SPSS v. 21.

## 3. Results

### 3.1. Patient Characteristics

Of the 350 patients who were invited to participate in the study, 47 (13.4%) did not agree to participate and 35 (10%) were unable to answer the survey. We finally included 268 (76.6%) patients.

Table 1 shows the main sociodemographic and clinical variables of the included patients. More than half (52.6%) were men and the mean age was 64.1 years (SD 12.4; min 25, max 91). Most participants (88.4%) lived with family, 17.9% had no schooling, and 56.7% were retired at the time of the study. Most patients attended the oncology department (*n* = 163, 60.8%). The main diagnoses were chronic leukaemia (lymphocytic or myeloid) (*n* = 48; 18.2%), breast or ovarian cancer (*n* = 48; 18.2%) and multiple myeloma (*n* = 42, 15.9%). The mean time since diagnosis was 60.3 months (SD 53.3) and most patients had stage IV cancer (*n* = 127, 83.6%). Chemotherapy was palliative in 82.8% of cases. Initial chemotherapy had been oral in 181 patients (67.8%), and the mean duration of current treatment was 12 months (SD 202).

### 3.2. Association of Patients’ Clinical and Sociodemographic Variables with Their Perception of the COVID-19 Pandemic

Table 1 shows that 16% of patients had experienced afraid when attending hospital, 4.1% had experienced a dangerous situation when attending hospital, 16.8% had felt they were receiving less medical care, and 30.2% preferred remote pharmaceutical care (always and almost always, in all cases).

Patients who said they had experienced fear when attending hospital (always or almost always) were more likely to live alone (9/43, 20.9% vs. 22/225, 9.8%, *p* = 0.04) and received lower number of daily doses (median 1, range 5.0 vs. median 2, range 9.5, *p* = 0.03) than those who experienced never, almost never or sometimes afraid.

Patients who said they had experienced a dangerous situation when attending hospital (always or almost always) were more likely to live alone than those who had never, almost never or sometimes had such an experience (4/11, 36.4% vs. 27/257, 10.5%, *p* = 0.009) and were more likely to attend the oncology department (10/11, 90.9% vs. 153/257, 59.5%, *p* = 0.04).

Patients who felt they had received less care because they were non-COVID patients (always or almost always), were less likely to attend the oncology department than the remaining patients (19/45, 42.2% vs. 144/223, 64.6%, *p* = 0.005).

Compared with patients who did not prefer remote pharmaceutical services (never, almost never or sometimes), those that did (almost or almost always) were more likely to have no schooling (21/81, 25.9% vs. 27/187, 14.4%, *p* = 0.046), to live alone (15/81, 18.5% vs. 16/187, 8.6%, *p* = 0.02), to have treatment in cycles (59/81, 72.8% vs. 99/187, 52.9%, *p* = 0.002) and to have suffered no adverse effects (45/81, 55.6% vs. 73/187, 39%, *p* = 0.01). These patients had been on oral chemotherapy for longer (mean 18, SD 191 vs. mean 14, SD 250, *p* = 0.02), and their current treatment for longer (mean 18, SD 189 vs. mean 10, SD 202, *p* = 0.002). We also found significant differences between the different hospitals in this aspect (*p* < 0.001).

In the multivariable analysis remote pharmaceutical services were less popular among patients who lived with their family compared to those who lived alone (aOR 0.29, 95% CI 0.13–0.71, *p* = 0.006) and among patients who had treatment in cycles compared to those on continuous treatment (aOR 0.37, 95% CI 0.19–0.72, *p* = 0.003).

### 3.3. Association of Different Variables with Adherence to Treatment

The number of patients classified as adherent was 252 (95.5%) according to prescription refill records. The mean value of the pill count was 92.8 (DS 12.5), and 183 patients (69.3%) had a count over 90%. Of the 252 patients classified as adherent by the refill records, 77 (30.6%) had a pill count below 90%.

The patients classified as adherent by the refill records were less likely to have perceived a dangerous situation when attending the hospital than those classified as non-adherent (9/255, 3.5% vs. 2/12, 16.7%, *p* = 0.025) and were less likely to prefer telemedicine (71/255, 72.2% vs. 9/12, 75%, *p* < 0.001).

Table 2 shows the variables associated with being adherent according to pill count, which we selected for the comparisons as it was the most conservative of the two classification methods in our study. Adherent patients were more likely than nonadherent patients to be women (94/183, 51.4% vs. 30/81, 37.0%, *p* = 0.03). Non-adherent patients were more likely to have felt afraid when attending hospital (always or almost always) than adherent patients (19/81, 23.5% vs. 24/183, 13.1%, *p* = 0.04). We also found significant differences between the different hospitals in this aspect (*p* < 0.001).

The multivariate analysis confirmed greater adherence among women than among men (aOR 2.11, 95% CI 1.20–3.73, *p* = 0.01). Patients who said they had felt afraid when attending the hospital (always or almost always) were less likely to adhere to treatment (aOR 0.47, 95% CI 0.23–0.96, *p* = 0.04) (Table 3).

## 4. Discussion

This study provides the first evaluation in clinical practice of adherence to orally administered antineoplastics and associated factors during the COVID-19 pandemic. More than 15% of the included patients said they had experienced fear when attending hospital, 5% had experienced a dangerous situation when attending hospital, nearly 17% felt they had received less care as a result of the COVID-19 pandemic, and 30% preferred telepharmacy. Factors associated with feeling afraid and preferring telepharmacy were living situation (with family or alone) and hospital characteristics. Although adherence, measured through pill count, was high overall (nearly 70% of the patients were considered adherent) it was lower in patients who said they had experienced fear when attending hospital.

While most patients in our study said they always or almost always preferred face-to-face to remote pharmaceutical care, previous studies have shown greater patient satisfaction with telemedicine services, which include both medical and pharmaceutical care, [12,22]. However, these studies were carried out during the first wave of the pandemic, whereas ours was conducted a year after the pandemic began, when knowledge and preventive measures against SARS-CoV-2 infection had increased.

Living alone versus living with family was associated with greater fear when attending hospital, having felt in danger when attending hospital, and preferring telepharmacy to in-person care. Previous studies identified high levels of stress in cancer patients during the pandemic, mainly due to feeling alone because of the precautionary restrictions put in place [23,24]. For this reason, health promotion interventions must be implemented to help vulnerable patients such as those with cancer, who may feel alone in similar situations.

To the best of our knowledge, this is the first study to assess the relationship between fear during hospital visits, as experienced by many patients during the COVID-19 pandemic, and adherence to orally administered antineoplastics (measured by pill count and by refill records). A recent study assessed how safety measures adopted during the first outbreak of COVID-19 affected patient anxiety levels and adherence to clinical visits and treatment plans [11]. The authors found that 81% of patients attended their scheduled blood tests and 73% went for scheduled imaging tests as part of their clinical follow-up. In our study, adherence to antineoplastic treatment was high, as in previous studies [17], but patients who had experienced fear when attending hospital showed lower adherence. One previous study assessing adherence to intravenous chemotherapy during the COVID-19 pandemic showed that a feeling of fear was the third most common reason for postponing treatment [12]. Other authors have also described the relationship between fear and adherence to chemotherapy in cancer patients [25,26], but focused on intravenous chemotherapy (measuring the mean number of days that hospital visits were postponed).

In our study, unlike in certain previous studies on chronic diseases, women were more adherent to treatment than men [27,28]. This association was independent of fear regarding hospital visits. In recent studies [29,30], women showed higher levels of stress and anxiety than men during the COVID-19 pandemic, but in our study, feelings of fear or danger in relation to the COVID-19 pandemic were experienced by both sexes in almost equal measure.

The pandemic has highlighted the growing need for chronic illness self-management strategies aimed at keeping the disease under control, minimising its impact on physical health and psychological sequelae [31]. Telemedicine provides an important benefit for immunosuppressed patients, such as cancer patients, as it helps to reduce the risk of SARS-CoV-2 infection [9]. Moreover, given that the COVID-19 pandemic is ongoing, and in view of predictions by infectious disease experts of more frequent and more deadly future pandemics [32], it is necessary to establish a strategy that encompasses different aspects of chronic disease management, including follow-up of adherence to medication. Patients’ health status could benefit from telemedicine services that include individual follow-up of adherence to treatment, among other aspects. Individualised health care must also take into account each patient’s social environment, as patients that live alone **could** have more difficulty attending hospital appointments. Including local health care professionals and community pharmacists in the hospital team would facilitate the integrated management of patients with no social support. In addition, given the differences between the participating hospitals, a shared protocol should be established for use in different settings.

Our study has some limitations. Firstly, because the COVID-19 pandemic began relatively recently, no previously validated questionnaires were available to assess patient perceptions based on existing restrictions, as has been shown previously [11]. Nevertheless, the questionnaire was developed by a group of experts through a Delphi procedure. Secondly, patients who have had COVID-19 may be more susceptible to fear and anxiety, but we were unable to access data on SARS-CoV-2 infection in our study sample. Nevertheless, since the prevalence of infection in our region during the study period was below 10% [33], we believe this factor is very unlikely to have influenced our results. Thirdly, recruitment was performed when patients came to the hospital to collect their medication, which could have led to selection bias if this had left out from to the study to those patients who were undergoing telepharmacy. However, the study was carried out in 2021, when practically all patients on orally administered antineoplastics went to the hospital in person. Finally, the vaccination of these patients could have influenced on the results; nevertheless, the vaccination program started in May 2021, so we do not consider it relevant to include this variable.

The strengths of our study include the large sample size and consecutive inclusion of patients, with a low frequency of refusal to participate, which helped to ensure the representativeness of our results. In addition, by reviewing medical records as well as conducting telephone surveys, we reduced possible biases in patient data collection. Lastly, we used two different methods to assess adherence (refill records and pill count), and as result, we were able to increase the precision of our main outcome variable by adopting the pill count classification over the less conservative refill records.

## 5. Conclusions

In conclusion, according to our results, the fear experienced by some patients when attending hospital during the COVID-19 pandemic has affected adherence to orally administered antineoplastics. Patients’ social environment (living alone versus with family) affects whether they experienced fear when attending hospital during the pandemic, or whether they preferred telepharmacy. Developing a unified care plan that takes into account each patient’s individual and social situation could improve patient management and adherence to treatment.

## Figures and Tables

**Table 1 jcm-11-02436-t001:** Relationship of patients’ sociodemographic and clinical characteristics with their perceptions during the COVID-19 pandemic.

Variable ^a^	All Patients (*n* = 268)	Have You Felt Afraid When Attending Hospital?			Have You Experienced Any Dangerous Situations When Attending Hospital?			Have You Felt That Non-COVID Patients Have Received Less Care?			Do You Prefer Remote Pharmaceutical Care?		
		No ^b^	Yes ^c^	*p* Value	No ^b^	Yes ^c^	*p* Value	No ^b^	Yes ^c^	*p* Value	No ^b^	Yes ^c^	*p* Value
**All patients (*n* = 268)**		225 (84.0)	43 (16.0)		257 (95.9)	11 (4.1)		223 (83.2)	45 (16.8)		187 (69.8)	81 (30.2)	
**Sociodemographic characteristics**													
**Sex**				0.23			0.63			0.28			0.92
Man	141 (52.6)	122 (54.2)	19 (44.2)		136 (52.9)	5 (45.5)		114 (51.1)	27 (60.0)		98 (52.4)	43 (53.1)	
Woman	127 (47.4)	103 (45.8)	24 (55.8)		121 (47.1)	6 (54.5)		109 (48.9)	18 (40.0)		89 (47.6)	38 (46.9)	
**Age (median, range)**	64.6 (65.7)	64.5 (65.7)	66.9 (46.2)	0.51	64.8 (65.7)	58.1 (35.1)	0.07	64.9 (65.7)	60.7 (45.9)	0.33	62.9 (65.7)	68.3 (53.5)	0.06
**Educational attainment**				0.56			0.74			0.75			0.046
No schooling	48 (17.9)	38 (16.9)	10 (23.3)		47 (18.3)	1 (9.1)		41 (18.4)	7 (15.6)		27 (14.4)	21 (25.9)	
Primary education	153 (57.1)	129 (57.3)	24 (55.8)		146 (56.8)	7 (63.6)		125 (56.1)	28 (62.6)		108 (57.8)	45 (55.6)	
Secondary or University education	67 (25)	46 (28.0)	9 (20.9)					57 (25.6)	10 (22.2)		52 (27.8)	15 (18.5)	
**Living situation**				0.04			0.009			0.92			0.02
Alone	31 (11.6)	22 (9.8)	9 (20.9)		27 (10.5)	4 (36.4)		26 (11.7)	5 (11.1)		16 (8.6)	15 (18.5)	
With family	237 (88.4)	203 (90.2)	34 (79.1)		230 (89.5)	7 (63.6)		197 (88.3)	40 (88.9)		171 (914.0)	66 (81.5)	
**Work status**				0.67			0.29			0.19			0.22
Unemployed	46 (17.2)	40 (17.8)	6 (14.0)		44 (17.1)	2 (18.2)		41 (18.4)	5 (11.1)		36 (19.3)	10 (12.3)	
In work	70 (26.1)	60 (26.7)	10 (23.3)		65 (25.3)	5 (45.5)		53 (23.8)	17 (37.8)		51 (27.3)	19 (23.5)	
Retired	152 (56.7)	125 (55.6)	27 (62.8)					129 (57.8)	23 (51.1)		100 (53.5)	52 (64.2)	
**Clinical characteristics**													
**Hospital**				0.19			0.85			0.71			<0.001
Alicante	51 (19.0)	46 (20.4)	5 (11.6)		48 (18.7)	3 (27.3)		42 (18.8)	9 (20)		45 (24.1)	6 (7.4)	
Elche	43 (16.0)	39 (17.3)	4 (9.3)		42 (16.3)	1 (9.1)		34 (15.2)	9 (20)		29 (15.5)	14 (17.3)	
Elda	117 (43.7)	93 (41.3)	24 (55.8)		112 (43.6)	5 (45.5)		97 (43.5)	20 (44.4)		67 (35.8)	50 (61.7)	
San Juan	57 (21.3)	47 (20.9)	10 (23.3)		55 (21.4)	2 (18.2)		50 (22.4)	7 (15.6)		46 (24.6)	11 (13.6)	
**Department**				0.08			0.04			0.005			0.37
Oncology	163 (60.8)	142 (63.1)	21 (48.8)		153 (59.5)	10 (90.9)		144 (64.6)	19 (42.2)		117 (62.6)	46 (56.8)	
Haematology	105 (39.2)	83 (36.9)	22 (51.2)		104 (40.5)	1 (9.1)		79 (35.4)	26 (57.8)		70 (37.5)	35 (43.2)	
Diagnosis				0.85			0.09			0.08			0.08
Lung adenocarcinoma	19 (7.2)	18 (8.0)	2 (4.7)		17 (6.6)	3 (27.3)		16 (7.2)	4 (8.9)		15 (8)	5 (6.2)	
Colorectal cancer	24 (9.1)	22 (9.8)	3 (7.0)		23 (8.9)	2 (18.2)		21 (9.4)	4 (8.9)		20 (10.7)	5 (6.2)	
Breast/ovarian cancer	48 (18.2)	43 (19.1)	7 (16.3)		47 (18.3)	3 (27.3)		44 (19.7)	6 (13.3)		38 (20.3)	12 (14.8)	
Prostate cancer	26 (9.8)	23 (10.2)	3 (7.0)		26 (10.1)	0		23 (10.3)	3 (6.7)		14 (7.5)	12 (4.8)	
CLL/CML	48 (18.2)	39 (17.3)	9 (20.9)		48 (18.7)	0		33 (14.8)	15 (33.3)		27 (14.4)	21 (25.9)	
Multiple myeloma	42 (15.9)	33 (14.7)	9 (20.9)		41 (16)	1 (9.1)		34 (15.2)	8 (17.8)		33 (17.6)	9 (11.1)	
Other	57 (21.6)	47 (20.9)	10 (23.3)		55 (21.4)	2 (18.2)					40 (21.4)	17 (21)	
**Disease stage**				0.48			0.23			0.48			0.85
1–3	25 (16.4)	21 (15.7)	4 (22.2)		22 (15.5)	3 (30.0)		21 (15.7)	4 (22.2)		18 (16.8)	7 (15.6)	
4	127 (83.6)	113 (84.3)	14 (77.8)		120 (84.5)	7 (70.0)		113 (84.3)	14 (77.8)		89 (83.2)	38 (84.4)	
**ECOG Performance Status**				0.11			0.005			0.31			0.09
0	69 (48.6)	63 (51.2)	6 (31.6)		69 (51.5)	0		64 (50.0)	5 (35.7)		44 (44.0)	25 (59.5)	
Other	73 (51.4)	60 (48.8)	13 (68.4)		65 (48.5)	8 (100.0)		64 (50.0)	9 (64.3)		56 (56.0)	17 (40.5)	
**Time since diagnosis, months (median, range)**	43.9 (339.4)	45.3 (232.7)	45.3 (339.4)	>0.99	45.3 (339.4)	22.6 (69.4)	0.22	43.8 (339.4)	46.7 (254.8)	>0.99	40.1 (339.0)	52.3 (250.8)	0.29
**Treatment**													
**Type of medication**				0.43			0.35			0.25			0.002
Continuous	158 (59)	135 (60.0)	23 (53.5)		153 (59.5)	5 (45.5)		128 (57.4)	30 (66.7)		99 (52.9)	59 (72.8)	
In cycles	110 (41)	90 (40.0)	20 (46.5)		104 (40.5)	6 (54.5)		95 (42.6)	15 (33.3)		88 (47.1)	22 (27.2)	
**Treatment objective**				0.54			0.92			0.16			0.50
Adjuvant	46 (17.2)	40 (17.8)	6 (14.0)		44 (17.1)	2 (18.2)		35 (15.7)	13 (17.3)		34 (18.2)	12 (14.8)	
Palliative	222 (82.8)	185 (82.2)	37 (86.0)		213 (82.9)	9 (81.8)		188 (84.3)	34 (75.6)		153 (81.8)	69 (85.2)	
**Number of daily doses**	2 (9.5)	2 (9.5)	1 (5.0)	0.03	2 (9.5)	2 (5.0)	0.50	2 (9.5)	1 (5.0)	0.53	2 (7.5)	2 (9.0)	0.36
**Adverse effects**				0.72			0.25			0.09			0.01
No	118 (44.0)	98 (43.6)	20 (46.5)		115 (44.7)	3 (27.3)		93 (41.7)	25 (55.6)		73 (39.0)	45 (55.6)	
Yes	150 (56.0)	127 (56.4)	23 (53.5)		142 (55.3)	8 (72.7)		130 (58.3)	20 (44.4)		114 (61.0)	36 (44.4)	
**First chemotherapy**				0.76			0.34			0.60			0.23
Oral	181 (67.8)	151 (67.4)	30 (69.8)		175 (68.4)	6 (54.5)		149 (67.1)	32 (71.1)		131 (70.1)	50 (62.5)	
Intravenous	86 (32.2)	73 (32.6)	13 (30.2)		81 (31.6)	5 (45.5)		73 (32.9)	13 (28.9)		56 (29.9)	30 (37.5)	
**Duration of oral CTx, months (mean, SD)**	14 (202)	15 (202)	12 (119)	0.40	14 (202)	9 (46)	0.94	14 (191)	16 (250)	0.51	14 (250)	18 (191)	0.02
**Duration of current CTx, months (mean, SD)**	12 (202)	12 (202)	14 (81)	0.51	12 (201)	17 (46)	0.44	11 (191)	18 (200)	0.10	10 (202)	18 (189)	0.002
**Total duration of CTx, months (mean, SD)**	14 (250)	14 (250)	12 (148)	0.47	14 (250)	7 (45)	0.63	14 (191)	16 (250)	0.66	14 (250)	18 (191)	0.42

^a^ All values are expressed as *n* (%) unless otherwise specified. ^b^ never, almost never or sometimes. ^c^ always or almost always. CLL: chronic lymphocytic leukaemia; CML: chronic myeloid leukaemia; CTx: chemotherapy; ECOG: Eastern Cooperative Oncology Group.

**Table 2 jcm-11-02436-t002:** Variables associated with adherence to treatment (pill count > 90%) (*n* = 264).

Variables ^a^	Adherent	Non-Adherent	*p* Value
**Sociodemographic characteristics**			
**Sex**			0.03
Man	89 (48.6)	51 (63)	
Woman	94 (51.4)	30 (37)	
**Age (median, range)**	65.5 (65.7)	65.4 (53.8)	0.59
**Educational attainment**			0.63
No schooling	31 (16.9)	16 (19.8)	
Primary education	104 (56.8)	48 (59.3)	
Secondary or University education	48 (26.2)	17 (21)	
**Living situation**			0.30
Alone	19 (10.4)	12 (14.8)	
With family	164 (89.6)	69 (85.2)	
**Work status**			0.47
Unemployed	30 (16.4)	16 (19.8)	
In work	51 (27.9)	17 (21)	
Retired	102 (55.7)	48 (59.3)	
**Clinical characteristics**			
**Hospital**			<0.001
Alicante	30 (16.4)	21 (25.9)	
Elche	39 (21.3)	4 (4.9)	
Elda	69 (37.7)	45 (55.6)	
San Juan	45 (24.6)	11 (13.6)	
**Department**			0.55
Oncology	108 (59.0)	51 (63.0)	
Haematology	75 (41.0)	30 (37.0)	
**Diagnosis**			0.22
Lung adenocarcinoma	13 (7.1)	6 (7.4)	
Colorectal cancer	15 (8.2)	9 (11.1)	
Breast and ovarian cancer	38 (20.8)	10 (12.3)	
Prostate cancer	17 (9.3)	9 (11.1)	
Chronic leukaemia (lymphocytic or myeloid)	32 (17.5)	16 (19.8)	
Multiple myeloma	34 (18.6)	8 (9.9)	
Other	34 (18.6)	23 (28.4)	
**Disease stage**			0.45
1–3	19 (18.4)	6 (13.3)	
4	84 (81.6)	39 (86.7)	
**ECOG Performance Status Scale**			0.49
0	48 (50.5)	19 (44.2)	
Other	47 (49.5)	24 (55.8)	
**Months since diagnosis (median, range)**	45 (338.6)	41 (237.0)	0.59
**Treatment**			
**Type of medication**			0.10
Continuous	102 (55.7)	54 (66.7)	
In cycles	81 (44.3)	27 (33.3)	
**Treatment objective**			0.32
Adjuvant	34 (18.6)	11 (13.6)	
Palliative	149 (81.4)	70 (86.4)	
**Number of daily doses**	2 (9.5)	2 (5.0)	0.99
**Adverse effects**			0.27
No	77 (42.1)	40 (49.4)	
Yes	106 (57.9)	41 (50.6)	
**First chemotherapy**			0.52
Oral	127 (69.4)	53 (65.4)	
Intravenous	56 (30.6)	28 (34.6)	
**Duration of oral chemotherapy, months (mean, SD)**	14 (202)	17 (136)	0.49
**Duration of current chemotherapy, months (mean, SD)**	12 (201)	11 (102)	0.76
**Total duration of chemotherapy, months (mean, SD)**	14 (250)	14 (167)	0.77
**Have you felt afraid when attending hospital?**			0.04
never + almost never + sometimes	159 (86.9)	62 (76.5)	
always + almost always	24 (13.1)	19 (23.5)	
**Have you experienced any dangerous situations when attending hospital**?			0.68
never + almost never + sometimes	176 (96.2)	77 (95.1)	
always + almost always	7 (3.8)	4 (4.9)	
**Have you felt that non-COVID patients have received less care?**			0.31
never + almost never + sometimes	156 (85.2)	65 (80.2)	
always + almost always	27 (14.8)	16 (19.8)	
**Do you prefer remote pharmaceutical care?**			0.23
never + almost never + sometimes	133 (72.7)	53 (65.4)	
always + almost always	50 (27.3)	28 (34.6)	

^a^ All values are expressed as *n* (%) unless otherwise specified.

**Table 3 jcm-11-02436-t003:** Multivariate analysis of associations between different variables and adherence to treatment (pill count > 90%) (*n* = 264).

Variable	ORa ^a^	95% CI	*p* Value
**Hospital**			
Alicante	1.00		
Elche	8.06	2.44–26.57	0.001
Elda	1.24	0.62–2.50	0.54
San Juan	3.16	1.31–7.66	0.01
**Sex**			
Man	1.00		
Woman	2.11	1.20–3.73	0.01
**Have you felt afraid when attending hospital?**			
never + almost never + sometimes	1.00		
always + almost always	0.47	0.23–0.96	0.04

^a^ Adjusted for the variables in the table.

## Data Availability

All the available data are included in this article.

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
