# Peer review of "Influence of the COVID-19 Pandemic on Adherence to Orally Administered Antineoplastics"

_jcm, 2022, doi:10.3390/jcm11092436_

Round 1

Reviewer 1 Report

SUMMARY:
Talens et al. presented their survey and questionary analysis evaluating the adherence to oral chemotherapy and the existence of fear during COVID-19 pandemic. This investigation is relevant and contribute to the knowledge of COVID-19 pandemic and possible consequences on vulnerable patients diagnosed with cancer. Unfortunately, the study design described may affect conclusions obtained by the authors.

Overall comments

Study Design

Selection method. The recruitment of participants was designed “in person” by approaching patients in the hospital. This approach indirectly selected patients who attended the hospital and left behind patients that were under remote clinical visits, telemedicine or those who had not receiving their oral chemotherapy. Another strategy would have been to select participants randomly from the hospitals data base, after invitation follow by enrollment a patient self-report method for adherence evaluation would have been used, while this may be outside the scope of this study, a comment about it would be beneficial in the discussion section.

Introduction

Line 47-49. Authors should review the statements presented in the manuscript, because Sarid et. al. work (Ref. 12) is focused on anxiety” (“adoption of precautionary measures in the hemato-oncology care setting can lead to a significant reduction in COVID-19-related anxiety”), while Karacin et. al. (Ref. 11) study “fear” and COVID-19.

Materials and Methods

Please provide information about the ethical approval for the study, indicating committee and registration number of approved investigation.

Line 97. Include the reference for the information presented (“73% and 81% of patients…”)

Lines 103-104. Indicate the months of study instead of “the first half of 2021”

Variables. For the instruments used to collect the primary and secondary outcome, would the authors include supplementary information with details about the validated survey and questionary used.

Reviewer 2 Report

In this study, the authors aim to evaluate the impact of fear and anxiety related to the COVID-19 pandemic among patients with cancer and its influence on adherence to systemic anti-cancer drugs. They analyze a patient population from four hospitals in Spain, and find that a substantial proportion of patients preferred telepharmacy, adherence according to prescription refill records was acceptable but not optimal, and that patients who were afraid were less adherent to anti-cancer treatment.

Overall, this is an interesting study that helps to better understand the patterns of adherence to therapy among patients with cancer and their relation to fear and anxiety from COVID-19.

However, there are some issues that remain to be addressed by the authors:

1-The authors state that "to date, no published studies have assessed the association between the feelings of fear and/or anxiety and adherence to oral antineoplastic treatment during the COVID-19 pandemic". Other similar statements are also found in the Discussion section. These statements are false, as different studies have assessed the fear and anxiety due to COVID-19 among patients with cancer, and some evaluated its influence on patient adherence to systemic therapies (e.g. Karacin C. et al., Future Oncology, 2020). This should be removed and the authors should discuss previous studies describing the fear and anxiety related to COVID-19 in patients with cancer in the Introduction or Discussion sections of the manuscript.

2-Regarding sample size calculation, could the authors provide more details how this was precisely achieved? What is the "previous study" being used as a "benchmark" to estimate the necessary sample size?

3-Did the authors only look at patients treated with cytotoxic chemotherapy, or did they also include patients treated with other types of anti-cancer agents, such as targeted therapy or immunotherapy?

4-As this study was conducted in 2021, at a time when COVID-19 vaccines started to be available, it would be important to know which patients were vaccinated at the time of the questionnaire, as this could influence their fear and anxiety in relation to COVID-19 and also their adherence to therapy.

5-In the Statistical Analysis paragraph, the authors state the following: "We estimated the prevalence of adherence to treatment (primary outcome) and the existence of fear, measured through the structured questionnaire (secondary outcomes), as well as the prevalence according to relevant variables". What is meant by "as well as the prevalence"? 

6-While it is true that the mortality rates in relation to COVID-19 infection is higher among patients with cancer as compared to the general population, the mortality rates identified for patients with cancer vary across studies, depending on the specific characteristics of the patients evaluated (i.e. hematological malignancies, advanced vs. early stage disease). Therefore, it might be appropriate to reference multiple studies and avoid using within the text a specific number for the mortality rate, as it is currently found in the Introduction.

7-In the Introduction section, it is important to highlight the fact that oncological communities (e.g. ASCO, ESMo, etc...) aimed to prioritize telemedicine over in-person encounters, only when (1) this is appropriate (i.e. no need to administer therapies or undergo imaging follow-ups), and (2) is technically feasible.

Round 2

Reviewer 1 Report

This reviewer appreciate the work invested to overcome the comments. The authors improved their manuscript and this reviewer has no further suggestions.

Thank you